# Socio-economic inequalities in the breadth of internet use before and during the COVID-19 pandemic among older adults in England

Olivia S. Malkowski[1], Nick P. Townsend[2], Mark J. Kelson[3], Charlie E. M. Foster[2], Max J. Western[1]*

1 Department for Health, Centre for Motivation and Health Behaviour Change, University of Bath, Bath, United Kingdom, 2 Centre for Exercise, Nutrition and Health Sciences, School for Policy Studies, University of Bristol, Bristol, United Kingdom, 3 Department of Mathematics, Institute of Data Science and Artificial Intelligence, University of Exeter, Exeter, United Kingdom

* M.J.Western@bath.ac.uk

**Data Availability Statement:** The data underlying the results presented in the study (SN 5050 and SN 8688) are available from the UK Data Service (https://ukdataservice.ac.uk/). Details on how to

## Abstract

Understanding digital exclusion in older adults during the COVID-19 pandemic could help tailor responses to future outbreaks. This cohort study used data from older adults aged 60+ years in England who participated in wave nine (2018/2019) of the main English Longitudinal Study of Ageing (ELSA) survey, and/or wave one of the ELSA COVID-19 sub-study (June/July 2020). Using latent class analysis and latent transition analysis, we aimed to identify distinct subgroups of older adults characterised by different patterns of internet use pre- and intra-pandemic, explore the extent to which individuals remained in the same subgroup or transitioned to a different subgroup during the COVID-19 pandemic, and examine longitudinal associations of socio-economic factors (education, occupational class, and wealth) with latent class membership. Preliminary tests showed that the types of internet activities differed between men and women; therefore, subsequent analyses were stratified by biological sex. Three clusters (low, medium, and high) were identified in male participants at both timepoints. Among female participants, three clusters were distinguished pre-pandemic and two (low versus high) during the pandemic. The latent classes were characterised by participants' breadth of internet use. Higher education, occupational class, and wealth were associated with greater odds of membership in the medium and/or high classes, versus the low class, in men and women. A high degree of stability in latent class membership was observed over time. However, men experienced a stark decrease in online health information-seeking. Our results highlight that inequality regarding the range of functional and social opportunities provided by the internet prevailed during the pandemic. Policymakers should ensure that digital access and upskilling initiatives are equitable for all.

## Introduction

The United Kingdom (UK) is an example of a 'super-aged' country that is forecasting over one-quarter of its population to be aged over 65 years by 2066 [1]. Recent internet use among

access the data, including the conditions of use, can be found on the ELSA website (https://www.elsa-project.ac.uk/accessing-elsa-data) and the UK Data Service website. The syntax files to replicate the analyses presented in this paper are openly available from GitHub, at https://github.com/OliviaMalkowski/Inequalities-Internet-COVID-19.git.

**Funding:** This work was supported by an Economic and Social Research Council (https://www.ukri.org/councils/esrc/) funded studentship through the South West Doctoral Training Partnership [grant number ES/P000630/1 to O.S. M.]. The funders had no role in study design, data collection and analysis, decision to publish, or preparation of the manuscript.

**Competing interests:** The authors have declared that no competing interests exist.

older adults in the UK aged 65 to 74 years has risen from approximately 52% to 83% between 2011 and 2019; of all adults aged 75 years and over, 20% were recent internet users in 2011, compared with 47% in 2019 [2]. Internet technology has the potential to enhance healthy ageing [3], social connectedness [4], and cognitive functioning [5]. However, older adults succumb to an age-related digital divide [6,7]. Indeed, while the older adult population has experienced the fastest increase in internet usage over recent decades [8], their engagement remains lower than other age groups [7,9].

Older adults are stereotypically described as a technology-resistant group of individuals. Nevertheless, older adults are diverse in the range of online activities they partake in [10,11]. As more older adults integrate the internet into their daily lives, understanding disparities in the use of digital media is becoming increasingly important [9]. Recent systematic reviews noted lack of interest, poor functionality, mistrust, accessibility issues, and inadequate support amongst the most important barriers to older adults' continued use of internet technology [12,13]. Hence, there is a need to ensure that older adults have the skills, resources, and technical assistance to navigate the digital landscape and maintain online safety [14].

The coronavirus (COVID-19) pandemic has disrupted the day-to-day lives of people in unprecedented ways, with profound impacts on the psychological and social well-being of populations globally [15,16]. While all age groups have been affected, older adults are disproportionately susceptible to severe outcomes of symptomatic infection from COVID-19, such as hospitalisation, the development of new conditions, and death [17–19]. Moreover, social isolation and loneliness were considered major risk factors for poor physical and mental health among older adults following the implementation of physical distancing and lockdown measures [20,21]. The vital role of, and the projected increased reliance on, the internet during and beyond the COVID-19 pandemic may disadvantage older citizens who are offline or not conversant in contemporary technologies [9,22]. Numerous studies have shown that older adults are able and willing to acquire digital literacy [10,23]. However, there is concern that a perceived lack of effort to engage older adults in digitisation during the COVID-19 pandemic could exacerbate inequalities [14,20].

Although most work to date has focused on correlates and outcomes of internet access, research delving into the specific activities for which the internet is used among older adults provides some evidence of differential uptake of behaviours relative to younger age groups [24]. Arguably, investigating the diversity of internet activities offers a more nuanced understanding of online participation and different types of digital exclusion in older adults, ensuring that we move beyond simply evaluating inequalities in access, and towards use and benefit. Past research has, at times, homogenised older adults' digital engagement; to keep up with the fast-paced digitalisation of all life domains, deconstructing variations in older adults' online activities is essential to inform policies supporting the expansion of digital skills. The literature has shown that older adults engage in a broad array of digital practices, to fulfil specific goals [25,26]. In one study [27], the authors identified four distinct subgroups of older adults based on the diversity of their online activities using latent class analysis (LCA). The clusters included a 'practical' group, characterised by respondents who used the internet primarily for functional purposes, including information-seeking and banking, a 'minimisers' group reporting the lowest frequency across most internet activities, a 'maximisers' group, who demonstrated the greatest frequency and diversity of internet use, and a 'social' group, whose members mainly used the internet for entertainment and social networking. Despite this important contribution to the literature, there remains a lack of longitudinal research exploring whether older adults shift from one latent class to another over time, particularly in response to unprecedented global events such as the COVID-19 pandemic, which accelerated digitalisation. A recent article employing LCA to identify participants' mobile phone usage styles indicated that

older adults increased the set of functions they adopted between 2016 and 2020, illustrated by a relatively high proportion (25%) of participants moving from the "limited" or "average" usage clusters, characterised by more traditional features, such as messaging/emails, to the "intensive usage" cluster, characterised by advanced usage across the full range of functions examined [26]. Although trends towards increasing usage diversification were apparent pre-pandemic, there was some evidence of an intensification of this process after the COVID-19 outbreak, which merits further attention, particularly as patterns of smartphone use may not mirror those of internet use more generally [26].

Common indicators of socio-economic status, such as education and income, are consistently and positively associated with internet use versus non-use [28], as well as the frequency of internet use among older adults who are online [29,30]. The literature also shows that people from varying socio-economic backgrounds use the internet for different activities [24,31]. Notably, greater breadth of internet use has been reported among more socio-economically advantaged older adults [32]. A recent study explored differences in COVID-19–related internet usage patterns among adults in the Netherlands [33]. Education, but not economic resources, assessed via participants' annual family income in the last 12 months, was positively associated with information and communication uses [33]. Previous work has shown that income is positively correlated with internet use [26,28], and may be particularly important in explaining occupational and consumptive activities [34], which were not assessed. The findings also reinforce the value of using multiple indicators of socio-economic status; this approach could help to elucidate the various dimensions' interrelationships and account for their common and distinctive associations with online practices. To explore the role of socio-economic status as comprehensively as possible, three objective indicators will be used in the present study: education, occupational class, and wealth. As a contemporary marker of socio-economic status, wealth is considered a more appropriate measure for older adults and could provide unique insights relative to education and occupational class, which are often established in early adulthood but do affect the life-cycle accumulation of wealth [35]. Of utmost relevance to researchers developing interventions to mitigate digital exclusion is a deeper understanding of the influence of the COVID-19 pandemic on widening or reducing complex inequalities across a broader range of online activities and digital devices, a gap which the present study seeks to address.

The aim of this study was to: (1) identify whether latent subpopulations, characterised by different patterns of internet use, exist in a sample of older adults in England before and during the COVID-19 pandemic; (2) use latent transition analysis (LTA) to explore the extent to which individuals remained in the same subpopulation or transitioned to a different latent class during the pandemic; and (3) examine associations of socio-economic variables (education, occupational class, and wealth) with latent class membership. No hypotheses were made regarding the number, characteristics, or evolution of the latent classes over time, as this analysis was deemed exploratory.

## Materials and methods

### Study design and participants

Data, accessed for research purposes on the 17[th] of December 2021, were drawn from wave nine (2018/2019) of the main English Longitudinal Study of Ageing (ELSA) survey as a 'pre-pandemic' baseline assessment [36], and wave one of the ELSA COVID-19 sub-study (June/July 2020) as a follow-up [37]. The authors did not have access to personal data.

ELSA is an ongoing, longitudinal survey, established in 2002. Data are collected biannually from a nationally representative sample of adults aged 50+ years living in private households

in England. The original respondents were recruited from households who participated in the Health Survey for England in 1998, 1999, or 2001. The sample is refreshed periodically to maintain the complete 50+ years age profile. All interviews have consisted of a face-to-face computer-assisted personal interview and a self-completion questionnaire [38]. Further information on the cohort is available elsewhere [39]

The COVID-19 sub-study was conducted on participants selected from the existing ELSA sample in the context of the COVID-19 outbreak. Participants completed the survey online (82%) or via computer-assisted telephone interviews (18%). The survey was issued to 9,525 study participants (7,689 issued core members), and 7,040 survey interviews (5,825 productive core members) were completed (~75% response rate) [40]. Details about the protocol for the COVID-19 sub-study can be found online [40].

Only core members aged 60+ years were included in the present work, to align with the World Health Organization's definition of older age [41]. ELSA core members met the following eligibility criteria: (1) fit the age criteria of a given ELSA cohort; (2) took part in the sample-origin Health Survey for England; and (3) were interviewed at the first opportunity when invited to join the study [38]. Ethical approval for wave nine of the main ELSA survey was granted by the South Central–Berkshire Research Committee [17/SC/0588], and by the University College London Research Ethics Committee for the COVID-19 sub-study. Participants provided written informed consent if they completed the survey online and oral informed consent if they completed it by telephone. The current study was approved by the Research Ethics Approval Committee for Health [EP 20/21 109] at the University of Bath. This study followed the Strengthening the Reporting of Observational Studies in Epidemiology (STROBE) reporting guidelines [42]; the checklist defines recommendations for what should be included in an accurate and complete report of an observational study (S1 Appendix).

## Measures

**Frequency and purpose of internet use.** The frequency of internet use was assessed at baseline by asking participants: 'On average, how often do you use the internet or email?'. In the COVID-19 sub-study, participants were asked how often they had used the internet or email since the COVID-19 outbreak. To account for coding discrepancies between waves (S1 Table), four categories were created: (1) High (More than once a day/Every day, or almost every day); (2) Moderate (At least once a week, but not every day); (3) Low (At least once a month, but not every week/Less than monthly/At least once every three months/Less than every three months); and (4) Never.

To assess the purpose of internet use among older adults who were online, respondents were asked 'For which of the following activities did you use the internet in the last 3 months?'. There were sixteen response options at baseline and twelve in the COVID-19 sub-study. Participants could indicate more than one answer. Eight categories (see S2 Table for a complete list of response options) of internet activities were created to ensure consistency across time-points: (1) Emails; (2) Calls; (3) Health (finding health-related information); (4) Entertainment; (5) News; (6) Market (online shopping); (7) Social networking; and (8) Internet transactions. Participants were assigned a value of 1 if they used the internet for at least one response option in the respective category, and 0 if they did not. Respondents who reported never using the internet (~19%) were excluded from analyses, as the response options representing different purposes of internet use were not applicable to them.

**Covariates.** Socio-economic status was assessed via three proxy measures, retrieved at baseline: education, occupational class, and wealth [35]. Education was measured according to the highest educational qualification obtained by participants and recoded into three

categories: (1) Low (no qualifications); (2) Medium (school qualifications); and (3) High (at least some tertiary education). Participants who selected 'foreign/other' (~8%) were treated as missing cases, as they could not be assigned to any of the educational categories generated for the present study [43]. Occupational class, based on participants' current or most recent occupation, was assessed using the three-class National Statistics Socio-Economic Classification [35]. Therefore, participants who had never worked and were long-term unemployed (~1%) were excluded from analyses. Wealth was operationalised as quintiles of total non-pension wealth at the benefit unit level [35]. Age data, obtained at baseline, were fed-forward to the COVID-19 sub-study [40].

## Statistical analyses

Preliminary analyses explored differences between men and women across the eight types of internet activities, using Pearson's chi-squared ($\chi^2$) test (S3 and S4 Tables). Given the significant differences found at both timepoints across six of the eight activities, all subsequent statistical analyses were performed separately for male and female participants.

First, we used LCA, a person-centred statistical method, to identify distinct subgroups of respondents who shared similar patterns of internet use, independently at baseline and follow-up. We estimated one- to six-class unconditional LCA models at each timepoint, using the eight categories of online activities as latent class indicators. The models were performed with 200 random starts and 50 final-stage optimisations. The model selection criteria included the Akaike Information Criterion (AIC), the Bayesian Information Criterion (BIC), and the sample-size adjusted BIC (SSABIC), where lower values indicated a better model fit. Furthermore, we used the Vuong-Lo-Mendell-Rubin likelihood ratio test and the Lo-Mendell-Rubin adjusted likelihood ratio test [44], which examine whether the fit of the current model is significantly better ($p<0.05$) than a model with one fewer class. Entropy, a measure of classification accuracy, was also inspected, with higher values (closer to one) indicating a more parsimonious solution. Moreover, the size of the smallest latent class was reviewed. The final models were selected according to fit indices, balanced with the interpretability of the latent classes.

After determining the optimal number of classes at each timepoint, descriptive labels were assigned to each class based on item-probability plots, displaying the conditional probabilities of endorsing the latent class indicators. Descriptive statistics were calculated for the overall sample at both timepoints, and separately for each class, as counts (*n*) and proportions (%) or mean (standard deviation). To validate the latent classes emerging from the unconditional models, we included age (a continuous variable), and three socio-economic factors (i.e., education, occupational class, and wealth, recoded as dummy variables), as explanatory variables of class membership in a multinomial logistic regression model. The LCA models with covariates were assigned fixed class-specific item probabilities, using values from the final unconditional LCA model at the respective timepoint [45,46]. As a preliminary exploration, changes in class membership from baseline to follow-up were examined using cross-sectional results from each wave. Participants were allocated to one of the classes emerging from the LCA models at each timepoint, using modal class assignment based on the posterior probabilities. Cross-classification matrices were produced to summarise changes among latent classes over time [45].

In the third step, the assumption of longitudinal measurement invariance (i.e., the structure of the latent classes is the same at both timepoints) was tested by comparing a full non-invariance model with freely estimated item probabilities at each timepoint to a full measurement invariance model where item probabilities were constrained to be equal over time [45]. Then, we performed unconditional LTA, an extension of the LCA framework for longitudinal designs, to examine transition patterns from pre- to intra-pandemic, among the 2,063 men

and 2,538 women with data at both timepoints. In the final step, we added covariates into the LTA models. Specifically, multinomial logistic regression predicted latent class membership at baseline based on age, education, occupational class, and wealth. For the unconditional and conditional LTA models, we extracted transition matrices describing the probability of participants changing class at follow-up, conditional on baseline class assignment. We did not explore associations between covariates and transition probabilities due to the high degree of stability observed in latent class membership over time (see results). All LTA models were run with 1,000 random starts, 250 final-stage optimisations, and a maximum of 20 iterations in the initial stage.

LCA and LTA models were performed using the maximum likelihood with robust standard errors estimator. Full-information maximum likelihood handled missing data in the unconditional models under the missing at random assumption, whereas listwise deletion occurred in models with covariates. Stata/BE Version 17.0 (College Station, TX: StataCorp LP) was used for data preparation, descriptive analyses, and producing cross-classification matrices. Mplus 8.7 [47] was employed to run the LCA, measurement invariance, and LTA models. Statistical significance was defined as $p < 0.05$. The Stata and Mplus syntax files are openly available at https://github.com/OliviaMalkowski/Inequalities-Internet-COVID-19.git.

## Results

### Cross-sectional LCA models

S5 Table displays the fit indices for each LCA before and during the COVID-19 pandemic. For men, the three-class solution was deemed optimal at each wave, as these models had the lowest BIC values. Furthermore, the Vuong-Lo-Mendell-Rubin likelihood ratio test and the Lo-Mendell-Rubin adjusted likelihood ratio test suggested there was no significant improvement in model fit for the four-class solution relative to the three-class solution at each timepoint (all $p > 0.05$).

For female participants, there was conflicting information for the three- to four-class solutions based on the fit indices at the pre-pandemic wave. Although the BIC fit indices indicated the four-class solution was a more parsimonious fit to the data, the likelihood ratio tests indicated no statistically significant improvement in model fit versus the three-class solution (both $p > 0.05$). Upon inspection of the probability plots, the three-class solution was chosen as the inclusion of a fourth latent class did not provide any additional meaningful information. At follow-up, the BIC fit indices favoured the four-class solution, whereas the likelihood ratio tests indicated the two-class solution was the most appropriate model for the data. Considering there were only small differences in the BIC and SSABIC values between the two- to four-class models, the two-class solution was selected as it had the highest entropy (0.647), indicating better differentiation between classes. Furthermore, one of the classes emerging from the four-class solution was small, comprising approximately 9% of the sample.

The item-probability plots for men (Fig 1) reflected an ordered class pattern, indicating that the latent classes were defined by participants' breadth of internet use. Therefore, in the remainder of this paper, 'breadth' is defined in terms of the probability of endorsing all latent class indicators. The classes were broadly comparable at both timepoints, except for the probability of finding information on health-related issues, which was lower across all classes during the COVID-19 pandemic. For each wave of data, the smallest class, defined as 'Low', comprised participants who evidenced relatively low probabilities across all latent class indicators. Respondents' internet use in this group was mainly limited to sending or receiving emails. There were 405 (22.3%) participants in the low group pre-pandemic and 301 (17.2%) during the COVID-19 pandemic. The second class, labelled 'Medium', had 681 (37.4%) participants

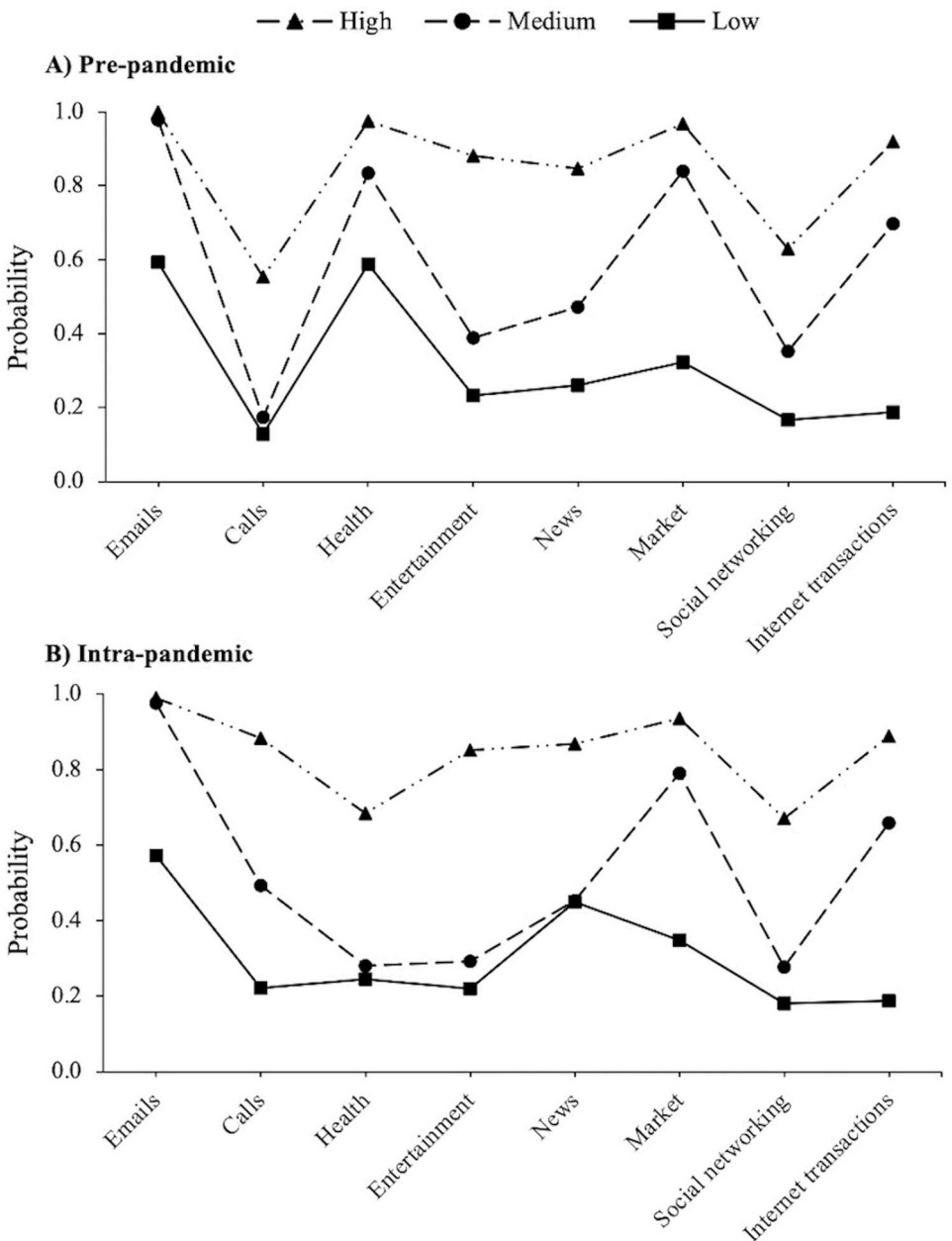

**Fig 1. Latent classes pre- (*n* = 1,819) and intra-pandemic (*n* = 1,750) by latent class indicators in male participants.**

pre-pandemic and 899 (51.4%) participants intra-pandemic. Members in the medium class had relatively low probabilities of reporting internet use for calls, entertainment, news, and social networking. Conversely, they displayed relatively high probabilities of using the internet for emails, online shopping, and internet transactions. The third class, named 'High', had 733 (40.3%) members at baseline and 550 (31.4%) at follow-up. Participants in the high class demonstrated the most diversity in reported internet activities at both timepoints. The plots also suggested that using the internet for video or voice calls increased to a greater degree among participants in the medium or high classes, relative to the low class, between the pre- and

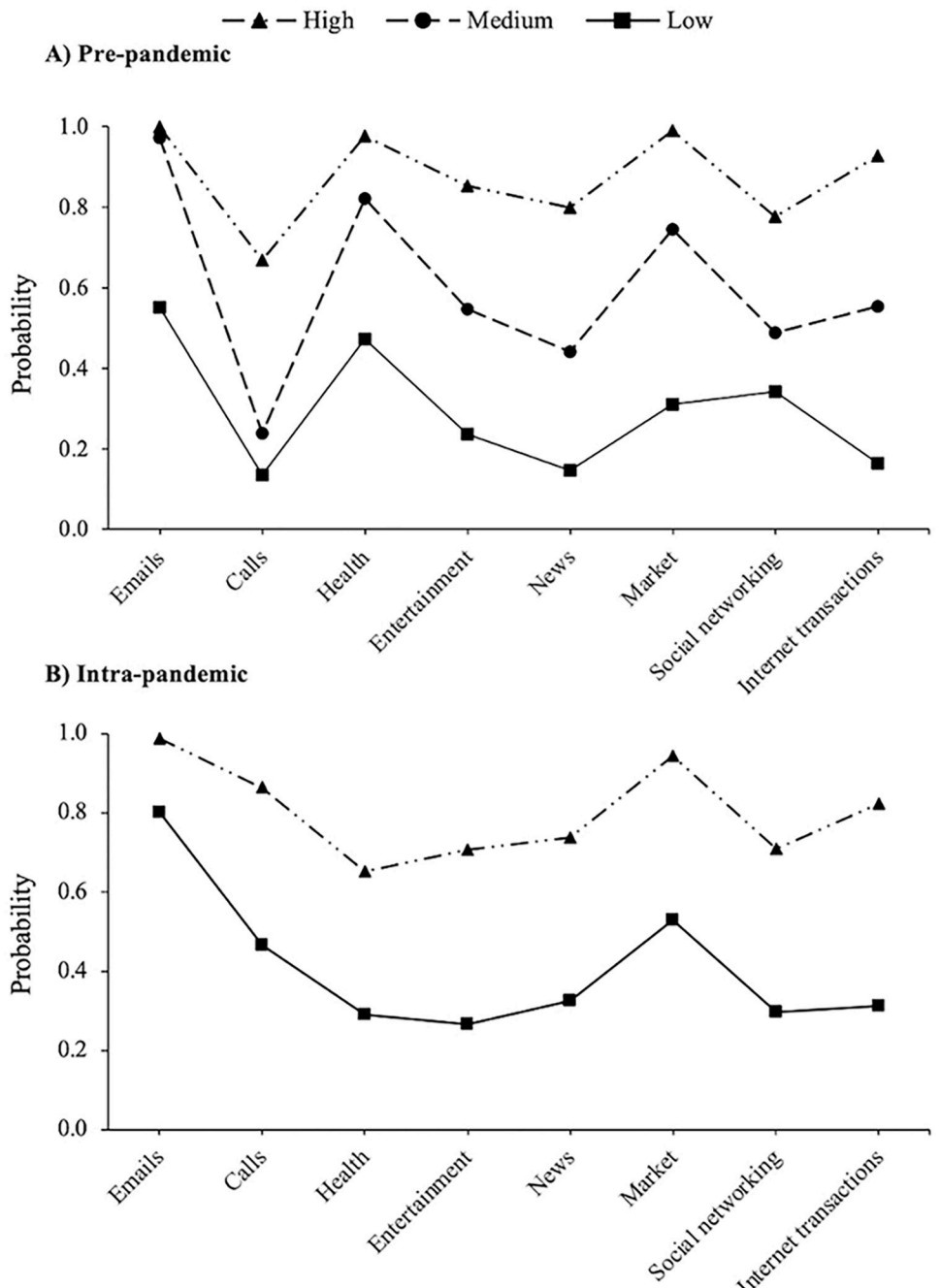

**Fig 2. Latent classes pre- (*n* = 2,235) and intra-pandemic (*n* = 2,158) by latent class indicators in female participants.**

intra-pandemic waves. Descriptive statistics for the total sample, and stratified by latent class membership, are available in S6 Table.

Fig 2 presents the item-probability plots for female participants. At the pre-pandemic wave, the first class, defined as 'Low', was comprised of 523 (23.4%) participants with relatively low probabilities across all latent class indicators. The largest class pre-pandemic, labelled 'Medium', had 1,247 (55.8%) participants. Members in this class had relatively low

probabilities of reporting internet use for calls, entertainment, news, social networking, and internet transactions. However, they displayed comparatively high probabilities of using the internet for emails, finding health-related information, and online shopping. The smallest class, named 'High', had 465 (20.8%) members at baseline. Participants in this group displayed the highest probabilities across all latent class indicators. At follow-up, two distinct classes emerged: participants in the first class, labelled 'Low' ($n$ = 1,274; 59.0%), used the internet mainly for sending or receiving emails, whereas those in the second class, defined as 'High' ($n$ = 884; 41.0%), demonstrated greater breadth of internet use. Descriptive statistics for the overall sample, and each latent class, are presented in S7 Table.

## Associations of socio-economic variables with latent class membership in the conditional LCA models

The logit coefficients and odds ratios from the multinomial logistic regression of the latent classes on socio-economic variables, adjusted for age, are shown in Table 1. For men, participants with medium or high education, versus low education, had significantly higher odds of membership in the medium group, whereas those with high education had significantly higher odds of being in the high group, relative to the low group (reference), at both waves (all $p<0.05$). Respondents in higher managerial, administrative, and professional occupations, compared with participants in routine and manual occupations, had significantly greater odds of membership in the medium or high classes, versus the low class, at baseline and follow-up (all $p<0.05$). At the pre-pandemic assessment, the fourth (Odds Ratio [OR] = 2.613, $p$ = 0.024) and fifth (OR = 7.800, $p<0.001$) quintiles of wealth were associated with significantly higher odds of membership in the high class, relative to the low group. Similar results were found during the COVID-19 pandemic, where being in the highest, relative to the lowest, quintile of wealth was associated with significantly higher odds (OR = 3.660, $p$ = 0.003) of membership in the medium group. Moreover, higher wealth (second to fifth quintiles versus first quintile) was associated with greater odds of being assigned to the high group, compared with the low group (all $p<0.05$).

In women, higher education (medium or high versus low), occupational class (intermediate or higher managerial, administrative, and professional occupations versus routine and manual occupations), and wealth (third to fifth quintiles versus first quintile) were associated with significantly increased odds of membership in the medium or high groups at baseline, and significantly higher odds of being assigned to the high group at follow-up, relative to the low class (reference) at each timepoint (all $p<0.05$).

## Cross-sectional transitions and measurement invariance

The cross-classification matrices describing movement among the internet classes over time, using the LCA results, are presented in S8 Table. For the male and female samples, full measurement invariance did not provide a good fit to the data, resulting in model non-identification. Consequently, we assumed full non-invariance of the internet classes across timepoints in the LTA models. The fit indices for the unconditional and conditional LTA models are shown in S9 Table.

## Latent transition probabilities extracted from the unconditional LTA models

The transition probabilities from the unconditional LTA models (Table 2) suggested a high degree of stability in latent class status. For male participants, the stability in patterns of

**Table 1. Multinomial logistic regression coefficients and odds ratios for the latent class models pre- and intra-pandemic with age and socio-economic variables as covariates, using the low class as the reference group.**

| | Estimate (SE) | p | OR (95% CI) |
|---|---|---|---|
| **Male participants** | | | |
| *Pre-pandemic (n = 1,627)* | | | |
| Medium class (*n* = 636) | | | |
| Age | -0.041 (0.014) | **0.005** | 0.960 (0.933, 0.988) |
| Education | | | |
| Low (reference) | | | |
| Medium | 0.817 (0.336) | **0.015** | 2.263 (1.172, 4.369) |
| High | 1.376 (0.345) | **<0.001** | 3.958 (2.014, 7.779) |
| Occupational class | | | |
| Routine and manual (reference) | | | |
| Intermediate | 0.314 (0.259) | 0.226 | 1.368 (0.823, 2.275) |
| Higher managerial, administrative and professional | 0.532 (0.244) | **0.029** | 1.702 (1.054, 2.747) |
| Wealth | | | |
| 1st quintile (reference) | | | |
| 2nd quintile | -0.142 (0.372) | 0.704 | 0.868 (0.418, 1.800) |
| 3rd quintile | -0.287 (0.340) | 0.399 | 0.751 (0.385, 1.462) |
| 4th quintile | -0.233 (0.351) | 0.506 | 0.792 (0.398, 1.575) |
| 5th quintile (highest) | 0.705 (0.394) | 0.073 | 2.024 (0.936, 4.377) |
| High class (*n* = 619) | | | |
| Age | -0.178 (0.018) | **<0.001** | 0.837 (0.808, 0.866) |
| Education | | | |
| Low (reference) | | | |
| Medium | 0.436 (0.408) | 0.286 | 1.546 (0.695, 3.440) |
| High | 1.505 (0.404) | **<0.001** | 4.505 (2.040, 9.945) |
| Occupational class | | | |
| Routine and manual (reference) | | | |
| Intermediate | 0.184 (0.280) | 0.510 | 1.203 (0.695, 2.080) |
| Higher managerial, administrative and professional | 1.192 (0.248) | **<0.001** | 3.293 (2.027, 5.351) |
| Wealth | | | |
| 1st quintile (reference) | | | |
| 2nd quintile | 0.205 (0.497) | 0.680 | 1.228 (0.464, 3.252) |
| 3rd quintile | 0.429 (0.438) | 0.327 | 1.536 (0.651, 3.624) |
| 4th quintile | 0.961 (0.426) | **0.024** | 2.613 (1.134, 6.021) |
| 5th quintile (highest) | 2.054 (0.458) | **<0.001** | 7.800 (3.180, 19.134) |
| *During COVID-19 (n = 1,478)* | | | |
| Medium class (*n* = 764) | | | |
| Age | -0.044 (0.018) | **0.015** | 0.957 (0.924, 0.992) |
| Education | | | |
| Low (reference) | | | |
| Medium | 1.185 (0.429) | **0.006** | 3.272 (1.411, 7.586) |
| High | 2.117 (0.438) | **<0.001** | 8.306 (3.517, 19.614) |
| Occupational class | | | |
| Routine and manual (reference) | | | |
| Intermediate | -0.377 (0.282) | 0.182 | 0.686 (0.395, 1.193) |
| Higher managerial, administrative and professional | 0.690 (0.276) | **0.012** | 1.994 (1.160, 3.427) |
| Wealth | | | |

*(Continued)*

**Table 1.** (*Continued*)

| | Estimate (SE) | p | OR (95% CI) |
|---|---|---|---|
| 1st quintile (reference) | | | |
| 2nd quintile | 0.528 (0.446) | 0.237 | 1.696 (0.707, 4.069) |
| 3rd quintile | 0.413 (0.401) | 0.303 | 1.512 (0.688, 3.319) |
| 4th quintile | 0.347 (0.389) | 0.373 | 1.414 (0.660, 3.033) |
| 5th quintile (highest) | 1.298 (0.441) | **0.003** | 3.660 (1.543, 8.682) |
| High class (*n* = 426) | | | |
| Age | -0.178 (0.021) | **<0.001** | 0.837 (0.804, 0.872) |
| Education | | | |
| Low (reference) | | | |
| Medium | 0.329 (0.422) | 0.436 | 1.389 (0.607, 3.180) |
| High | 1.585 (0.418) | **<0.001** | 4.881 (2.152, 11.070) |
| Occupational class | | | |
| Routine and manual (reference) | | | |
| Intermediate | 0.557 (0.319) | 0.081 | 1.746 (0.934, 3.263) |
| Higher managerial, administrative and professional | 1.709 (0.310) | **<0.001** | 5.524 (3.007, 10.149) |
| Wealth | | | |
| 1st quintile (reference) | | | |
| 2nd quintile | 1.534 (0.563) | **0.006** | 4.635 (1.537, 13.977) |
| 3rd quintile | 1.086 (0.516) | **0.035** | 2.961 (1.078, 8.138) |
| 4th quintile | 1.551 (0.492) | **0.002** | 4.717 (1.799, 12.367) |
| 5th quintile (highest) | 2.850 (0.525) | **<0.001** | 17.295 (6.184, 48.367) |
| **Female participants** | | | |
| *Pre-pandemic (n = 1,872)* | | | |
| Medium class (*n* = 959) | | | |
| Age | -0.101 (0.015) | **<0.001** | 0.904 (0.878, 0.931) |
| Education | | | |
| Low (reference) | | | |
| Medium | 0.922 (0.227) | **<0.001** | 2.514 (1.612, 3.920) |
| High | 1.067 (0.253) | **<0.001** | 2.906 (1.770, 4.773) |
| Occupational class | | | |
| Routine and manual (reference) | | | |
| Intermediate | 0.599 (0.207) | **0.004** | 1.821 (1.214, 2.730) |
| Higher managerial, administrative and professional | 1.233 (0.235) | **<0.001** | 3.433 (2.165, 5.444) |
| Wealth | | | |
| 1st quintile (reference) | | | |
| 2nd quintile | -0.088 (0.298) | 0.768 | 0.916 (0.511, 1.642) |
| 3rd quintile | 0.613 (0.295) | **0.038** | 1.846 (1.035, 3.290) |
| 4th quintile | 0.606 (0.284) | **0.033** | 1.833 (1.051, 3.198) |
| 5th quintile (highest) | 0.928 (0.306) | **0.002** | 2.529 (1.390, 4.604) |
| High class (*n* = 440) | | | |
| Age | -0.268 (0.024) | **<0.001** | 0.765 (0.730, 0.801) |
| Education | | | |
| Low (reference) | | | |
| Medium | 1.139 (0.383) | **0.003** | 3.122 (1.475, 6.608) |
| High | 1.803 (0.389) | **<0.001** | 6.065 (2.831, 12.991) |
| Occupational class | | | |
| Routine and manual (reference) | | | |

(*Continued*)

**Table 1.** (Continued)

| | Estimate (SE) | p | OR (95% CI) |
|---|---|---|---|
| Intermediate | 1.522 (0.299) | **<0.001** | 4.582 (2.551, 8.230) |
| Higher managerial, administrative and professional | 2.481 (0.315) | **<0.001** | 11.954 (6.446, 22.165) |
| Wealth | | | |
| 1$^{st}$ quintile (reference) | | | |
| 2$^{nd}$ quintile | -0.483 (0.502) | 0.336 | 0.617 (0.230, 1.651) |
| 3$^{rd}$ quintile | 1.221 (0.408) | **0.003** | 3.390 (1.523, 7.546) |
| 4$^{th}$ quintile | 1.091 (0.400) | **0.006** | 2.978 (1.359, 6.525) |
| 5$^{th}$ quintile (highest) | 1.951 (0.407) | **<0.001** | 7.034 (3.169, 15.614) |
| *During COVID-19 (n = 1,712)* | | | |
| High class (*n* = 729) | | | |
| Age | -0.129 (0.012) | **<0.001** | 0.879 (0.858, 0.900) |
| Education | | | |
| Low (reference) | | | |
| Medium | 0.688 (0.284) | **0.015** | 1.990 (1.140, 3.476) |
| High | 1.319 (0.295) | **<0.001** | 3.741 (2.097, 6.673) |
| Occupational class | | | |
| Routine and manual (reference) | | | |
| Intermediate | 0.467 (0.188) | **0.013** | 1.595 (1.103, 2.307) |
| Higher managerial, administrative and professional | 0.817 (0.202) | **<0.001** | 2.265 (1.525, 3.363) |
| Wealth | | | |
| 1$^{st}$ quintile (reference) | | | |
| 2$^{nd}$ quintile | -0.066 (0.322) | 0.839 | 0.936 (0.498, 1.760) |
| 3$^{rd}$ quintile | 0.750 (0.283) | **0.008** | 2.117 (1.215, 3.689) |
| 4$^{th}$ quintile | 0.786 (0.274) | **0.004** | 2.194 (1.283, 3.751) |
| 5$^{th}$ quintile (highest) | 1.271 (0.273) | **<0.001** | 3.563 (2.086, 6.084) |

*Note*: *SE*, standard error; *OR*, odds ratio; *CI*, confidence intervals. The counts (*n*) are based on participants' most likely latent class membership. Bold denotes statistical significance (*p*<0.05).

**Table 2.** Estimated latent transition probabilities in male and female participants' patterns of internet use from pre- to intra-pandemic (unconditional LTA models).

| | During COVID-19 | | |
|---|---|---|---|
| Pre-pandemic | Low class (34.3%) | Medium class (30.1%) | High class (35.7%) |
| *Male participants (n = 2,063)* | | | |
| Low class (34.3%) | 0.994 | 0.000 | 0.006 |
| Medium class (29.8%) | 0.000 | 1.000 | 0.000 |
| High class (35.9%) | 0.008 | 0.009 | 0.983 |
| | Low class (58.7%) | | High class (41.3%) |
| *Female participants (n = 2,538)* | | | |
| Low class (15.2%) | 1.000 | – | 0.000 |
| Medium class (42.9%) | 0.984 | – | 0.016 |
| High class (41.9%) | 0.007 | – | 0.993 |

*Note*: LTA, latent transition analysis. The proportions (%) are based on participants' most likely latent class pattern. The latent transition probabilities are based on the estimated model.

internet use was highest among those in the medium group, who had a probability of 1.000 of remaining in that group at follow-up. Participants in the low and high groups at baseline also displayed high probabilities (low: 0.994; high: 0.983) of staying in the same class during the COVID-19 pandemic. Of the participants in the low group at baseline, approximately 0.6% transitioned to the high group at follow-up. Among those in the high group pre-pandemic, 0.8% transitioned back to the low group, while 0.9% changed to the medium group at the intra-pandemic assessment.

For female participants, there were high probabilities that respondents in the low (1.000) or medium (0.984) classes at baseline would be assigned to the low group at the intra-pandemic wave. A considerable percentage (99.3%) of participants in the high class at baseline maintained their class membership at follow-up. Approximately 1.6% of women in the medium class pre-pandemic transitioned into the high class, while 0.7% of participants in the high group at baseline transitioned back to the low group during the COVID-19 pandemic. The latent transition probabilities for male and female participants were comparable in the conditional LTA models and are reported in S10 Table.

### Associations of socio-economic variables with pre-pandemic latent class membership in the conditional LTA models

The multinomial logistic regression logit coefficients and odds ratios from the conditional LTA models, with age, and socio-economic variables entered as covariates on baseline class membership are presented in S11 Table. Results were comparable to the conditional LCA models at the pre-pandemic wave. Notably, male participants with medium or high, versus low, educational qualifications had significantly higher odds of being in the medium group (all $p<0.01$), and those with high education had significantly greater odds (OR = 3.946, $p = 0.001$) of being in the high group, versus the low group (reference). Men in higher managerial, administrative, and professional occupations, compared with respondents in routine and manual occupations, had significantly increased odds of being in the medium (OR = 1.664, $p = 0.032$) or high (OR = 4.172, $p<0.001$) classes, relative to the low class. Participants in the fifth, versus the first, quintile of wealth had significantly higher odds (OR = 2.515, $p = 0.007$) of being assigned to the medium class, and those in the fourth (OR = 4.322, $p = 0.004$) or fifth (OR = 13.871, $p<0.001$) quintiles had significantly greater odds of membership in the high class, relative to the low class.

In female participants, higher education (medium or high versus low), and occupational class (intermediate or higher managerial, administrative, and professional occupations versus routine and manual occupations) were associated with greater odds of membership in the medium or high groups, versus the low group (all $p<0.01$). Participants in the third to fifth, versus the lowest, quintiles of wealth had significantly higher odds of being assigned to the high class (all $p<0.001$), and those in the fifth quintile (OR = 3.266, $p = 0.023$) had significantly greater odds of membership in the medium class, relative to the low class.

### Discussion

In this study, we sought to identify distinct clusters of older adults based on the online activities they engaged in. Differences in the types of internet activities used were observed by biological sex. In our male participant sample, we identified three latent classes before and during the COVID-19 pandemic. Participants in the low class used the internet mainly for traditional purposes, such as sending or receiving emails. Second, a medium class emerged, characterised by internet use for the purposes of accessing emails, online shopping, and internet transactions. Finally, participants in the high class displayed high probabilities of using the internet

for all online activities, including social networking and media. The high class (40.3%) was the most prevalent subgroup pre-pandemic, whereas the medium class (51.4%) had the most participants at follow-up. However, the probability of using the internet to find information on health-related issues was lower across all classes during the COVID-19 pandemic. At the pre-pandemic assessment, three classes were identified among women, reflecting an ordered pattern with increasing breadth of internet use. The medium class, comprising 55.8% of the sample at baseline, diverged slightly from the corresponding class in men, as members used the internet mainly for sending or receiving emails, finding health-related information, and online shopping. In the COVID-19 sub-study, a two-class model was selected as the most appropriate solution for the female participant sample, consisting of a low and a high group.

Higher education, occupational class, and wealth were consistently associated with higher odds of membership in the medium and/or high classes, relative to the low class, at both timepoints in men and women. These findings corroborate previous evidence showing that socioeconomic status is positively associated with both the type and breadth of internet activities among older adults [32,48]. The observation of three distinct patterns of internet use in men and two to three classes among women complements and extends prior work. The high and low classes agree with previously identified clusters in a representative sample of older adults in the Netherlands [27]. Interestingly, the authors identified two other classes: 'practical users', who reported internet use for activities such as information-seeking, comparing products, and banking; and 'social users', who mainly used the internet for leisure-related purposes or social networking. Although the practical cluster closely aligns with the medium classes emerging in the present work, we did not identify a social class in male or female participants at either timepoint.

The full non-invariance LTA models were retained for male and female participants, meaning that the probability of endorsing each latent class indicator within a given class differed between timepoints. Overall, a high degree of stability was observed in latent class membership over time, with fewer than 2% of participants transitioning from pre- to intra-pandemic. Although our results did not provide much support for an intensification or reduction of the digital divide during the COVID-19 pandemic, a more nuanced exploration of the qualitatively different subgroups highlighted important sex differences. Indeed, while a lower proportion of men and women used the internet for health information during the pandemic, relative to the pre-pandemic assessment, this decline was particularly marked in the male sample. This finding agrees with the results of a recent study, showing that women were more likely to engage with health-related online content during the pandemic, whereas men showed more interest in online discussions surrounding the societal, economic, and political circumstances [49]. One explanation is that older men may be at increased risk of anxiety from COVID-19 media coverage [50], and thus limited their consumption of health-related coverage. Alternatively, it is possible that older adults sought health information from more traditional sources during the COVID-19 pandemic, such as via communicating with family members and friends, or by watching television. The decreased use of the internet for health-related issues is consistent with the reduction in healthcare utilisation that occurred during the pandemic period [51], which may have been due to a higher perceived risk of hospital-based COVID-19 transmission in prospective patients for other diseases, and/or the pandemic-induced postponement of planned treatments and medical examinations. It could also be a result of message fatigue and a subsequent diminished inclination to look for health information, due to prolonged, repeated exposure to COVID-19 recommendations and preventive messaging [52]. Rigorous qualitative research investigating people's experiences of health-seeking behaviours beyond the COVID-19 pandemic will be important to document trends in the resurgence of these online activities.

Given that socio-economic measures, such as education and occupational class, are frequently determined in early life and have lasting impacts on digital engagement [35,53], there is a need to identify novel means of promoting digitisation in older adults [54], particularly for those of lower socio-economic status, to prevent them from being left behind in an increasingly digital world. Correspondingly, gaining a deeper understanding of the underlying mechanisms should be a high priority for scholars. For instance, a study showed that socio-economic disparities in the breadth of internet use were less pronounced when controlling for digital skills [32]. Although further analyses are needed to confirm the relationship between socio-economic status, digital literacy, and patterns of internet use, this work highlights digital skills training for older adults as a potential area of intervention. To establish whether this is a viable means of fostering digital inclusion, our findings support the need for sex-specific consideration of these associations.

The current study was strengthened by the large sample of older adults in the ELSA dataset, which allowed us to stratify analyses by biological sex. Furthermore, the longitudinal design enabled an exploration of patterns of stability and transition among latent classes over time. However, there are several limitations to acknowledge. Notably, this study relied on self-report measures, which are prone to recall and social desirability biases. We treated socio-economic factors as time-constant variables. However, these variables may have fluctuated during the COVID-19 pandemic [55]. As years of educational attainment were not available in ELSA, the variable representing participants' highest educational qualification was used instead, which precluded the inclusion of older adults with foreign qualifications whose educational levels were not specifically stated. In addition, a direct comparison of internet uses between timepoints was not possible, as the categorisation of online activities in the COVID-19 sub-study differed from previous waves [16]. The mode of data collection changed between the two timepoints; the survey was administered predominantly online in the COVID-19 sub-study, which could have disrupted the continuity of measurement. It is therefore difficult to distinguish whether differences in the latent classes across waves are due to circumstances tied to the COVID-19 pandemic, the mode switch and altered question response options, or a host of other contemporaneous factors, events, and trends that are beyond the scope of this manuscript. ELSA participants were predominantly White. Therefore, replication with minority ethnic and racial groups is warranted to improve external validity. Moreover, since we only adjusted for age and the three indicators of socio-economic status in the conditional LCA and LTA models, the potential influence of other covariates and residual confounding should be considered when interpreting the results.

Our results inform researchers and policymakers about socio-economic subgroups that may benefit from intervention programmes designed to facilitate engagement in a broad array of online activities [48]. Importantly, the current study showed that socio-economic inequalities in internet use, which were well-established pre-pandemic, remained during the COVID-19 pandemic. Digital inclusion is increasingly recognised as a social determinant of health [56]; if inequalities in internet use persist during extreme events such as the COVID-19 pandemic, there is concern these could lead to widening health disparities in the aftermath of this and other similar pandemics or lockdowns. Aligned with this standpoint, future work should consider the addition of distal outcomes (e.g., mental health) in the LTA modelling framework, given the potential positive and negative influences of certain online activities on older adults' health and well-being [16,27,57,58]. Moreover, we recommend replicating or extending this study over the upcoming years, to investigate generational differences in the prevalence of the identified clusters and estimated transition probabilities between latent classes [27]. Due to the acceleration and continuation of activities such as remote working, as well as the permanent closure of many high street shops and services post-pandemic, an important policy

implication of the finding that internet usage was stable before and during the COVID-19 pandemic is the threat to societal engagement among those who are digitally excluded. Few participants in the lower classes of internet use pre-pandemic transitioned to a higher latent class during the COVID-19 pandemic, which suggests there are barriers to internet use, across varying online activities, amongst some individuals that must be investigated and tackled, such that equitable health and social care is available to all in the present time and in the upcoming years as digital delivery surges. Nonetheless, it is also important to consider that the low cluster's internet engagement could be based on conscious decisions rather than barriers or deficiencies. To ensure the societal inclusion of all groups of older people, services and support must be accessible in both digital and traditional (e.g., face-to-face) formats.

Overall, this study suggests it is possible to identify meaningful subgroups of older adults based on simple indicators of internet use. Furthermore, we provide evidence for socio-economic disparities in the breadth of internet use among older adults in England, which persisted during the COVID-19 pandemic. Of note is the observation that male participants experienced a sharp decline in online health information-seeking during the pandemic, which merits further attention. As the digitalisation of healthcare and most other major services is increasing over time, there is a suggestion that the digital divide, as well as health and social inequalities, will become more prominent in the future. Subsequently, our findings reiterate the importance of deploying targeted interventions to support digital inclusion in older adults, tailored according to the needs of varying demographic and socio-economic groups.

## Supporting information

**S1 Appendix. STROBE checklist.** *Note*: *Give information separately for cases and controls in case-control studies and, if applicable, for exposed and unexposed groups in cohort and cross sectional studies. Information in bold in the "Recommendation" column has been added by the study authors to designate the sections of the manuscript that cover each respective recommendation from the STROBE guidelines. Checklist reference: von Elm E, Altman DG, Egger M, Pocock SJ, Gøtzsche PC, Vandenbroucke JP. The Strengthening the Reporting of Observational Studies in Epidemiology (STROBE) statement: guidelines for reporting observational studies. Prev Med. 2007 Oct;45(4):247–51.
(DOCX)

**S1 Table. Response options belonging to the categories representing different frequencies of internet use.** *Note*: *ELSA*, English Longitudinal Study of Ageing. Survey question at baseline: On average, how often do you use the internet or email? Survey question at follow-up: Since the coronavirus outbreak, on average, how often did you use the internet or email?
(DOCX)

**S2 Table. Response options belonging to the categories representing different types of internet use.** *Note*: *ELSA*, English Longitudinal Study of Ageing. Survey question at baseline and follow-up: For which of the following activities did you use the internet in the last 3 months?
(DOCX)

**S3 Table. Pre-pandemic differences in the online activities performed by male and female participants.** *Note*: *n*, number of participants. Bold denotes statistical significance ($p < 0.05$).
(DOCX)

**S4 Table. Intra-pandemic differences in the online activities performed by male and female participants.** *Note*: *n*, number of participants. Bold denotes statistical significance ($p<0.05$).
(DOCX)

**S5 Table. Fit indices for the pre- and intra-pandemic latent class solutions.** *Note*: *AIC*, Akaike Information Criterion; *BIC*, Bayesian Information Criterion; *SSABIC*, sample-size adjusted Bayesian Information Criterion; *VLMR-LRT p*, Vuong-Lo-Mendell-Rubin likelihood ratio test *p*-value; *Adj. LMR-LRT p*, Lo-Mendell-Rubin adjusted likelihood ratio test *p*-value. Bold font indicates the selected model. The class sizes are based on participants' most likely latent class membership.
(DOCX)

**S6 Table. Descriptive statistics for the total sample, and stratified by latent class, pre- and intra-pandemic (male participants).** *Note*: *n*, number of participants; *SD*, standard deviation. [a]Age was collapsed to 90 for participants aged 90+ years. Demographic information was collected at baseline and fed-forward or updated in the COVID-19 sub-study.
(DOCX)

**S7 Table. Descriptive statistics for the total sample, and stratified by latent class, pre- and intra-pandemic (female participants).** *Note*: *n*, number of participants; *SD*, standard deviation. [a]Age was collapsed to 90 for participants aged 90+ years. Demographic information was collected at baseline and fed-forward or updated in the COVID-19 sub-study.
(DOCX)

**S8 Table. Preliminary cross-classification (transition) tables based on cross-sectional LCA results.** *Note*: *LCA*, latent class analysis. The proportions (%) are based on participants' most likely latent class membership.
(DOCX)

**S9 Table. Fit indices for the unconditional and conditional LTA solutions.** *Note*: *LTA*, latent transition analysis; *AIC*, Akaike Information Criterion; *BIC*, Bayesian Information Criterion; *SSABIC*, sample-size adjusted Bayesian Information Criterion.
(DOCX)

**S10 Table. Estimated latent transition probabilities in male and female participants' patterns of internet use from pre- to intra-pandemic (conditional LTA models).** *Note*: *LTA*, latent transition analysis. The proportions (%) are based on participants' most likely latent class pattern. The latent transition probabilities are based on the estimated model.
(DOCX)

**S11 Table. Pre-pandemic multinomial logistic regression coefficients and odds ratios for the full non-invariance LTA models with age and socio-economic variables as covariates, using the low class as the reference group.** *Note*: *LTA*, latent transition analysis; *SE*, standard error; *OR*, odds ratio; *CI*, confidence intervals. The counts (*n*) are based on participants' most likely latent class pattern. Bold denotes statistical significance ($p<0.05$).
(DOCX)

## Author Contributions

**Conceptualization:** Olivia S. Malkowski.

**Data curation:** Olivia S. Malkowski.

**Formal analysis:** Olivia S. Malkowski.

**Funding acquisition:** Olivia S. Malkowski.

**Methodology:** Olivia S. Malkowski.

**Project administration:** Olivia S. Malkowski, Max J. Western.

**Resources:** Olivia S. Malkowski.

**Supervision:** Nick P. Townsend, Mark J. Kelson, Charlie E. M. Foster, Max J. Western.

**Validation:** Olivia S. Malkowski.

**Visualization:** Olivia S. Malkowski.

**Writing – original draft:** Olivia S. Malkowski.

**Writing – review & editing:** Olivia S. Malkowski, Nick P. Townsend, Mark J. Kelson, Charlie E. M. Foster, Max J. Western.

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
