## [Decision Letter · Decision Letter 0]

23 Jan 2024

PONE-D-23-33851Socio-economic inequalities in the breadth of internet use before and during the COVID-19 pandemic among older adults in EnglandPLOS ONE

Dear Dr. Western,

Thank you for submitting your manuscript to PLOS ONE. After careful consideration, we feel that it has merit but does not fully meet PLOS ONE’s publication criteria as it currently stands. Therefore, we invite you to submit a revised version of the manuscript that addresses the points raised during the review process.

We look forward to receiving your revised manuscript.

Kind regards,

Sandar Tin Tin

Academic Editor

PLOS ONE

Journal Requirements:

2. Thank you for stating the following in the Acknowledgments Section of your manuscript: "ELSA is funded by the National Institute on Aging (R01AG017644), and by UK Government Departments coordinated by the National Institute for Health and Care Research(NIHR)."

Please remove any funding-related text from the manuscript and let us know how you would like to update your Funding Statement. Currently, your Funding Statement reads as follows: "This work was supported by an Economic and Social Research Council (https://www.ukri.org/councils/esrc/) funded studentship through the South West Doctoral Training Partnership [grant number ES/P000630/1 to O.S.M.]. The funders had no role in study design, data collection and analysis, decision to publish, or preparation of the manuscript."

Reviewers' comments:

Reviewer's Responses to Questions

**Comments to the Author**

1. Is the manuscript technically sound, and do the data support the conclusions?

Reviewer #1: Yes

Reviewer #2: Yes

2. Has the statistical analysis been performed appropriately and rigorously? 

Reviewer #1: Yes

Reviewer #2: I Don't Know

3. Have the authors made all data underlying the findings in their manuscript fully available?

Reviewer #1: Yes

Reviewer #2: Yes

4. Is the manuscript presented in an intelligible fashion and written in standard English?

Reviewer #1: Yes

Reviewer #2: Yes

5. Review Comments to the Author

Reviewer #1: Thank you having the opportunity to read the intriguing manuscript. The topic of the research (social inequalities in internet use among older individuals) is highly relevant and research on that field is still evolving. The study is based on high quality data and applies a superior method. I very much appreciate the longitudinal perspective as well as the SROBE-statement. However, the manuscript could be improved.

Before I come to my suggestions for revision, I would like to point out that I am not an expert in the method used (LCA). However, I have tried to familiarize myself with the method and review the manuscript and the analyses to the best of my ability and conscience.

- The literature section needs to be revised. Please summarize the literature relevant for your argumentation and, based on that, elaborate your study design and research goals

- Some references do not seem to be well chosen.

- What role plays the COVID-19 pandemic in your study (accelerator for digitalization processes or impacts on health, impacts on economy,…)? Why are you interested in socio-economic outcomes of different purposes of internet use? And why do you use three distinct variables to depict the socio-economic situation? To what extend are they distinct from each other? Please strengthen your “story”/argumentation to underscore the relevance of your paper

- Why are the analyses conducted separated for men and women? Please provide a sound argumentation. Are you identifying similar sub-groups if you would have conducted analyses separated for men and women? The information that LCA is conducted separately for men and women pops up abruptly for the reader.

- Why is it important to increase the internet use for different purposes, as we know nothing about the reasons for comparatively lower use

- The analyses appear to have been carried out thoroughly and in accordance with existing standards

(Please find more detailled comments in the document attached.)

Reviewer #2: The article addresses an interesting topic. I can make a relevant contribution to PLOS ONE and the digital inequalities area. I agree that research looking at how different activities on the internet are related to different characteristics and factors in relevant to understand digital inequalities of older people. I miss the inclusion of more factors in the final analysis. For instance, we know that among the older population, age matter. The manuscript focuses on discussing self-reported biological sex, and the observed difference may be a consequence of other factors too (i.e. age, physical and mental health, ethnicity, etc.).

I miss some additional references to studies that have also looked at activities older people do online.

Fernández-Ardèvol, M., Rosales, A. and Cortès, F.M., 2023. Set in stone? Mobile practices evolution in later life. Media and Communication, 11(3), pp.40-52.

Beneito-Montagut, R., Rosales, A. and Fernández-Ardèvol, M., 2022. Emerging digital inequalities: A comparative study of older adults’ smartphone use. Social Media+ Society, 8(4), p.20563051221138756

6. PLOS authors have the option to publish the peer review history of their article (what does this mean?). If published, this will include your full peer review and any attached files.

Reviewer #1: **Yes: **Lisa Kortmann

Reviewer #2: No

---

## [Author Response · Author response to Decision Letter 0]

6 Mar 2024

See response to reviewers letter uploaded with submission.

---

## [Decision Letter · Decision Letter 1]

9 Apr 2024

PONE-D-23-33851R1Socio-economic inequalities in the breadth of internet use before and during the COVID-19 pandemic among older adults in EnglandPLOS ONE

Dear Dr. Western,

Thank you for submitting your manuscript to PLOS ONE. After careful consideration, we feel that it has merit but does not fully meet PLOS ONE’s publication criteria as it currently stands. Therefore, we invite you to submit a revised version of the manuscript that addresses the points raised during the review process.

We look forward to receiving your revised manuscript.

Kind regards,

Sandar Tin Tin

Academic Editor

PLOS ONE

Journal Requirements:

Reviewers' comments:

Reviewer's Responses to Questions

**Comments to the Author**

1. If the authors have adequately addressed your comments raised in a previous round of review and you feel that this manuscript is now acceptable for publication, you may indicate that here to bypass the “Comments to the Author” section, enter your conflict of interest statement in the “Confidential to Editor” section, and submit your "Accept" recommendation.

Reviewer #1: All comments have been addressed

2. Is the manuscript technically sound, and do the data support the conclusions?

Reviewer #1: Partly

3. Has the statistical analysis been performed appropriately and rigorously? 

Reviewer #1: Yes

4. Have the authors made all data underlying the findings in their manuscript fully available?

Reviewer #1: Yes

5. Is the manuscript presented in an intelligible fashion and written in standard English?

Reviewer #1: Yes

6. Review Comments to the Author

Reviewer #1: I am pleased that the feedback has been carefully reviewed and the manuscript revised. The manuscript has gained in strength and clarity.

However, I would like to make two points:

1. Please consider that the differences in latent classes may not have been evoked only by the COVID-19 pandemic or a mode effect. The study dates make it possible to speak of "pre-pandemic" and "during the COVID-19 pandemic", but the COVID pandemic can only be discussed as an explanation. (lines 516-518)

2. The authors very often use brackets to add additional information. If the information in the brackets is not essential, it can simply be deleted from the text. If the information is important, it does not have to be in brackets. For me as a reader, it is easier if the authors decide what is really relevant information in the main text and what is actually not so important and dispensable. In this way, the authors could streamline and shorten their text a little.

7. PLOS authors have the option to publish the peer review history of their article (what does this mean?). If published, this will include your full peer review and any attached files.

Reviewer #1: **Yes: **Lisa Kortmann

---

## [Author Response · Author response to Decision Letter 1]

18 Apr 2024

Dear Editor,

Thank you for the opportunity to revise the manuscript titled “Socio-economic inequalities in the breadth of internet use before and during the COVID-19 pandemic among older adults in England”. We appreciate the reviewer’s follow-up suggestions to improve the quality of the manuscript. To address feedback on the discussion section, we have reiterated that other explanations for differences in the latent classes over time may exist, beyond the COVID-19 pandemic and the switch in the mode of data collection; we have also used suggestive language throughout, so as not to infer causality. Furthermore, we have taken on board the reviewer’s feedback regarding the use of brackets, and have deleted unnecessary text and/or removed brackets where appropriate to streamline the manuscript.

Journal Requirements:

Author response: We have reviewed our reference list to ensure that it is complete and correct. We have also checked that none of the cited papers have been retracted. No changes have been made to the reference list during this round of revisions.

Point-by-point response to Reviewer 1’s comments:

Reviewer #1: I am pleased that the feedback has been carefully reviewed and the manuscript revised. The manuscript has gained in strength and clarity.

Author response a: We thank the reviewer for these encouraging comments.

However, I would like to make two points:

1. Please consider that the differences in latent classes may not have been evoked only by the COVID-19 pandemic or a mode effect. The study dates make it possible to speak of "pre-pandemic" and "during the COVID-19 pandemic", but the COVID pandemic can only be discussed as an explanation. (lines 516-518)

Author response b: Thank you for raising this. We have ensured that the discussion section reflects that the COVID-19 pandemic and the mode of data collection are not the only plausible explanations for differences in the latent classes over time. Correspondingly, we have made changes to lines 495-497 and 546-548.

2. The authors very often use brackets to add additional information. If the information in the brackets is not essential, it can simply be deleted from the text. If the information is important, it does not have to be in brackets. For me as a reader, it is easier if the authors decide what is really relevant information in the main text and what is actually not so important and dispensable. In this way, the authors could streamline and shorten their text a little.

Author response c: Thank you for this suggestion. Accordingly, we have either deleted the text if the information was not essential or removed the brackets if the information was deemed important. Changes have been made to the following lines: 51, 70-71, 94, 97, 112-114, 122, 124-126, 159, 161, 187-188, 229-230, 235, 245, 362, 373-374, 431, 437, 445, 457-458, 461, 467-468, 543-544, 577, 802-803, 822-823, and 827-828.

---

## [Editor Report · Decision Letter 2]

19 Apr 2024

Socio-economic inequalities in the breadth of internet use before and during the COVID-19 pandemic among older adults in England

PONE-D-23-33851R2

Dear Dr. Western,

We’re pleased to inform you that your manuscript has been judged scientifically suitable for publication and will be formally accepted for publication once it meets all outstanding technical requirements.

Kind regards,

Sandar Tin Tin

Academic Editor

PLOS ONE

---

## [Editor Report · Acceptance letter]

26 Apr 2024

PONE-D-23-33851R2 

PLOS ONE

Dear Dr. Western, 

I'm pleased to inform you that your manuscript has been deemed suitable for publication in PLOS ONE. Congratulations! Your manuscript is now being handed over to our production team.

Kind regards, 

on behalf of

Dr. Sandar Tin Tin 

Academic Editor

PLOS ONE